# Chemical, Textural and Antioxidant Properties of Oat-Fermented Beverages with Different Starter Lactic Acid Bacteria and Pectin

**DOI:** 10.3390/biotech13040038

**Published:** 2024-09-25

**Authors:** Dmitrii V. Khrundin, Elena V. Nikitina

**Affiliations:** Department of Meat and Milk Technology, Kazan National Research Technological University, 420015 Kazan, Russia; khrundin@yandex.ru

**Keywords:** oat-based beverage, lactic acid bacteria, fermentation, viscosity, texture, antioxidants

## Abstract

Currently, starter cultures for fermenting plant-based beverages are not widely available commercially, but producers can use starter cultures for dairy products. Therefore, the aim of this study was to determine the physicochemical, rheological, antioxidant and sensory properties of oat beverages with/without pectin fermented by four different dairy starter cultures. The use of a mono-starter with *Lactobacillus bulgaricus* or *Sreptococcus thermophilus* allows for the efficient use of glucose, and more lactic acid is accumulated. The beverage with *L. bulgaricus* is characterised by high adhesion, syneresis and low cohesiveness, and it has high antioxidant activity and a low sensory profile. Using starter with *L. bulgaricus*, *S. thermophilus* and some *Lactococcus* for fermentation yields a product with high sensory capacity, forming a high-viscosity beverage matrix with low syneresis, high water retention, chewy texture and stickiness. It has been observed that the absence of lactococci and the presence of *Lactobacillus casei*, *L. Rhamnosus* and *L. paracasei* in the starter yields a product with high antioxidant activity, especially in the presence of pectin. The use of pectin significantly improves the viscosity and textural properties of oat yoghurt, enhancing the drink’s flavour and giving it body. For many reasons, the use of different commercial starters in the dairy industry results in different viscosities of oat fermented beverages, forming a matrix with different textural, sensory and antioxidant properties.

## 1. Introduction

Plant-based food ingredients account for a significant share of current market conditions. For example, with an annual growth rate of 7.2%, the global market for plant proteins could reach $15.6 billion by 2026 [1]. Cereal products (wheat, rice, maize, rye, barley and oats) are an integral part of the diet of many people and are used to feed animals. It is still the case that about 60% of the energy of people in developing countries comes from cereal products [2]. Oats (*Avena sativa*) belong to the Poaceae family and are the sixth most important cereal crop (in terms of production). They are widely grown in many regions, including the European Union, Canada, Russia, Australia, the United States and Brazil [3]. Oats are the basis for a wide range of popular foods such as oatmeal, porridge, cereal bars, biscuits, pasta, bread, yogurt, oat milk and drinks [4,5,6,7]. The demand for products based on oats is due to their high protein content (around 12–20%), high digestibility (around 90–94%) and significant amounts of fibre (for example, around 4–8% beta-glucans) [8]. 

The benefits of consuming oat products in the prevention of metabolic diseases are determined by the presence of components such as β-glucan (soluble fibre), avenntramides, tocols, sterols, phytic acid and avenacosides in oats. A review article by Varma et al. (2016) [9] highlighted oats as an effective way to prevent non-communicable diseases such as diabetes, obesity, hypertension and cardiovascular diseases, obesity, type 2 diabetes, gastrointestinal disorders and cancer. The presence of β-glucans in oats allows for the inclusion of oats and products based on them in healthy functional foods; moreover, the beneficial properties of glucans have been proven in chronic diseases [3]. 

One of the oldest processes used by humans to preserve animal and plant products is fermentation [10]. In modern times, fermentation can be used not only for preservation but also to produce new foods with unique flavour, aroma or texture. For such products, cereal crops, whose diversity is very high, are well suited as raw materials. In addition, cereals are a treasure trove of polysaccharides that can be used as carbon and energy sources by microorganisms during fermentation. The most traditional group of bacteria used in food products are lactic acid bacteria. Based on the metabolic capabilities of lactic acid bacteria, many authors have shown that cereal crops are optimal substrates for the growth of lactic acid bacteria (LAB), including those with probiotic potential [11]. In particular, this group is made up of the extensive members of the *Lactobacillus*, *Leuconostoc*, *Pediococcus*, *Lactococcus* and *Streptococcus* [10]. A lot of authors show the possibilities of using lactic acid bacteria for fermentation of plant food: *L. plantarum* [12,13], *L. casei* [12,14,15], *L. paracasei* [16,17,18], *L. rhamnosus* [17,19], *L. fermentum* [20,21] and others. In particular, LAB (including *Lactobacillus*, *Bifidobacteria*, *Lactobacillus*, *Cheeseba*, *Streptococcus*, *Myxobacterium*, etc.) are the most commonly used strains for fermenting oat beverages in current studies, while yeast and *Candida* are the main fermenting fungi. Their role is not only to produce lactic acid, it should be noted that these strains improve the nutritional, digestibility and organoleptic properties of fermented oat beverages [22,23]. Plant-based beverages can be a matrix for the transfer of probiotic bacteria [24], especially since some of the lactobacilli are commonly known as probiotics [25]. There is increasing evidence from several authors that cellular components of probiotics and/or their metabolites interacting with host cells can induce probiotic effects [25,26,27,28,29]. Metabolites and cellular components of Lactobacillus species have advantages over probiotic bacteria of other genera [30,31]. 

At present, pectins are widely used as natural stabilising agents, prebiotics, emulsifying agents and thickening agents in fermented dairy products [32]. Pectin is a hydrocolloid, an anionic polysaccharide based on galacturonic acid residues. The target use of pectin in fermented foods is widening. Pectin can be a protective agent for lactic acid bacteria [33], function as an encapsulating agent [34], and inhibit undesirable reactions in food products [35]. The prebiotic function of pectin is well known [32], but the plant fermented beverage consisting of prebiotic pectin and probiotic LAB can claim to be called synbiotic. This means that its ingredients (nutrients and bacteria) will play a role in regulating the abundance and diversity of the gut microbiota, maintaining gut health and preventing chronic disease [36].

Although there is a variety of research on different lactobacilli for plant beverage fermentation applications, the prevalence is still low. Starter cultures for milk fermentation are more widely available on the world market, specific starters for plant beverages are not yet widespread. The availability of ready-made starter cultures is important for food processing plants. The aim of this study was to determine the physicochemical, rheological, antioxidant and sensory properties of oat fermentations produced using four types of starters, all of which are widely available on the Russian market and are used in the production of dairy products. In addition, the study shows the differences in the complex properties of the oat yoghurt when pectin was used in the formulation. This study will provide much needed knowledge on the formulation of plant fermented beverages and the effect of different ingredients (starters and pectins) on texture and flavour, which will help in the development of the next plant fermented beverages.

## 2. Materials and Methods

### 2.1. Oat Material, Starter Cultures and Samples

Oat flakes were purchased from a local superstore in Kazan city, Russia. Oats flakes were sown and soaked in the water for 12 h. Oat flakes were purchased from a local supermarket in Kazan, Russia. Oat flakes were sieved and soaked in water for 12 h (1:2 ratio of crushed oat flakes/water). The swollen flakes were then ground with water in a laboratory blender (JustBuy, Shangzhan, China) to a homogeneous mass. The mixture mass was hydrolysed at a temperature of 60 ± 2 °C for 40 min (monitoring the increase in glucose content) to improve sensory characteristics due to the partial degradation of starch and an increase in the content of free simple sugars using the amylolytic enzyme product (Alfalad™ BN, BioPreparat, Moscow, Russia). To stop hydrolysis, oat base was heated to 95–98 °C for 5 min to inactivate enzymes and increase the stability of the food system. Then, the suspension was quickly cooled to 37 °C and physicochemical parameters were determined. Next, starter cultures were added to the mixture.

Starter LABs (“Lactosynthesis”, Moscow, Russia) served as a commercial strain for milk dairy products. LABs were stored in de Man, Rogosa, and Sharpe (MRS) broth (Himedia, India) with 50% glycerol at −80 °C. For culture activation, a 100 μL aliquot of each culture was individually transferred into MRS broth or milk whey broth and incubated at 37 °C for 24 h. The description of the samples and starter culture is presented in the Table 1. 

### 2.2. Preparing of Starters and Fermentation of Oat Base

The starter bacteria were cultured from lyophilised powder, grown into de Man, Rogosa, and Sharpe (MRS) broth (Himedia, India) at 37 °C for 24 h to obtain the pre-culture. The pre-cultures of lactic acid bacteria were prepared by incubation at 37 °C for 8–12 h. Overnight grown cells of LAB were used as inoculum such that a cell concentration of 10^8^ colony-forming units (CFU)/mL could be attained at 3 mL in the fermented oat base. Fermentation was carried out in glass containers containing 100 mL of oat base and kept at 37 °C for 8–12 h. Pectin (commercial apple high methoxyl pectin (“Pectowin”, Przemysiu, Poland) in an amount of 1% was added to half of the samples. All samples were kept for stabilisation at 4 °C for 24 h in the refrigerator. The physicochemical, textural, and antioxidant properties were evaluated.

### 2.3. Chemical Assays

Titratable acidity, pH and glucose content were determined as previously described [37]. The quantitative analysis of protein, fat, solids and total sugars in the samples was performed using Infra LUM^®^ FT-12 equipment (Lumex, St. Petersburg, Russia) with the appropriate software and calibration data. 

### 2.4. Preparing Protein-Free Extract (PFE) and Total Phenolic Compound (TPC) Assays

#### 2.4.1. Preparing of Protein-Free Extract (PFE)

For protein-free extract (PFE), 5 mL of 1% trichloroacetic acid solution was added to 5 g of sample and mixed [38]. After 5 min of incubation at room temperature, the precipitate was removed by centrifugation for 15 min at 10,000× *g*. The supernatant was used as a PFE.

#### 2.4.2. Total Phenolic Compound (TPC) Assays

Fermented oat beverage (1 g) was mixed with 5 g distilled water. The resulting mixture was used to determine the total amount of phenolic compounds in the product. Total phenolic compounds (TPCs) in beverage and PFE were determined using Folin–Ciocalteu reagent (CDH, New Delhi, India) [38]. 

### 2.5. Isolation and Yield of Extractable Polysaccharides (EPSs)

Extraction and quantification of extractable polysaccharides were performed as described previously [38]. The yield of EPSs was determined by the phenol sulphuric method [39]. Glucose was used as a calibration standard.

### 2.6. Apparent Viscosity and Thixotropic Property Assays

Viscosity measurements were taken using a rotational viscometer model RV-DVIII (Ningbo, China). Approximately 50 mL of sample was placed in a beaker and the viscosity measurements were taken using spindle 2 of the viscometer at 30 rpm. The temperature of the samples was about 25 °C. The viscosity values were calculated automatically using a coefficient to convert the viscosity values into centipoise units (cP). Measurements were carried out in 3 replicates for each treatment and results were expressed in mPa·s. 

The stability of the structure and thixotropic properties of the samples were evaluated by the coefficient of loss of viscosity (Lη, %) and coefficient mechanical stability (CMR), according to equations:L_η = ((η_s − η_e))/η_s ∙100,
CMR = η_e/η_s,
where ηs is the initial viscosity of the undisturbed structure, cP; ηe is the viscosity of the maximally destroyed structure, cP.

### 2.7. Syneresis Analysis

Syneresis was measured after cooling samples weighing about 10 g to 4 °C after 24 h of storage. The samples were centrifuged for 5 min at 1000 rpm and a temperature of 20 °C. The released serum was removed and weighed. Syneresis (%) was calculated by the following equation:Syn = S/M∙100, (%), 
where M is the mass of the sample, g; S is the mass of the released serum, g.

### 2.8. Water-Holding Capacity Analysis

The water-holding capacity (WHC) was measured after cooling the samples weighing about 20 g to 4 °C after 24 h of storage. The samples were centrifuged for 10 min at 3000 rpm and a temperature of 20 °C. The released serum was removed and weighed. WHC (%) of the product was calculated by the following equation: WHC = (M − W)/M∙100,
where M is the sample weight, g; W is the released serum weight, g.

### 2.9. Texture Profile Analysis (TPA)

The texture profile analysis test was carried out using an ST-2 texture analyser (Quality Laboratory JSC, Moscow, Russia) as described previously [40]. The following factors were determined: hardness (g), adhesiveness, adhesiveness and gumminess [37]. 

### 2.10. Sensory Analysis

The sensory evaluation of the samples was carried out by an untrained non-smoking group of 20 subjects (10 men and 10 women aged 19 to 65 years). The samples were equilibrated to room temperature after storage for 24 h at 6 °C and a heaped teaspoon per person and sample was prepared on a small dish 30 min before the sensory test. Subjects were asked in advance if they had any food allergies, as oats have the potential to cause them. Ethical approval was also obtained for the publication of the results. Subjects were asked to evaluate the samples by appearance, colour, taste, smell, consistency and general acceptability. The sensory evaluation criteria are described in the Appendix A. A scaling of the intensity between 0 and 5 with 1 step was chosen with 0 being the weakest and 5 the strongest feature expression. 

### 2.11. Antioxidant Assays

#### 2.11.1. Ferric-Reducing Antioxidant Power Assay (FRAP)

The ferric-reducing antioxidant power (FRAP) analysis was analysed according to [41]. Two-fold pre-dilution beverage samples were used for the analysis.

#### 2.11.2. Evaluation of Radical-Scavenging Ability (RSA) by 2,2-Di-Phenyl-1-Picrylhydrazyl (DPPH) Assay

The radical-scavenging capacity was analysed according to [41]. Ten-fold pre-diluted beverage samples were used for analysis.

#### 2.11.3. OH Free-Radical-Scavenging Assay

OH free-radical-scavenging ability was carried out following the procedure described by Sungatullina et al. [41]. Ten-fold pre-diluted beverage samples were used for analysis.

### 2.12. Statistical Analysis

All the experiments were carried out in triplicate and replicated at least twice. Results are expressed as average ± standard deviation (SD). Data analysis and processing was carried out using Microsoft Excel 2016, GraphPad Prism8.0.1.244, Origin10 software. The values of *p* < 0.05 were considered statistically significant.

## 3. Results

### 3.1. Reasons for Choosing Raw Materials, Preparation Oat Flakes and Base

Oat flakes are obtained by processing natural oats. Oats contain vitamins (A, E, K, B) and minerals (K, Ca, Mg, P, Zn); rich in vegetable protein and soluble and insoluble fibres. Oat-based products have an important feature—the ability to envelop the walls of the stomach with a protective film and reduce the acidity of gastric juice, which is important for problems with the gastrointestinal tract such as gastritis, peptic ulcers and meteorism [42,43,44]. In addition, oat products have a specific and pleasant smell, sweet taste and an attractive colour. Thus, oat products can be used as raw materials to create new food products. The production of fermented oat base included the following stages and parameters (Figure 1). 

The first step in the production of oat flakes is the removal of impurities from the flakes. Then soaked at a temperature of 25.0 ± 3.0 °C for 8–12 h to swell proteins and polysaccharides, loosen the shells and release intermolecular bonds. The soaked flakes (after removal of the soak water) were subjected to wet grinding with a 1:2 hydromodule on a laboratory mill (Just Buy, China) at rotational speed of 36,000 rpm for 3 min to a homogeneous state (approximate particle size 50–300 Mesh).

The hydrolysis process practically did not affect the properties of the oat base. The content of dry substances and total sugar (glucose) increased as expected (Table 2). Some increase in protein and fat is probably due to their release from the native protein-fat-carbohydrate matrix under the action of an enzyme and heating. The standard process for obtaining protein from oats is heating with enzymatic treatment, which significantly increases the yield of protein components from plant raw materials [45]. Due to the short duration of treatment, our case shows a similar effect at a lower level.

The initial oat base had the characteristic flavour and aroma of raw cereals. After hydrolysis, the sensory features of the oat base improved significantly, the taste became more balanced with a pronounced sweetness.

### 3.2. The Oat Base Fermentation

#### 3.2.1. Chemical Composition

Oat base can be efficiently fermented by all the used starter cultures. In the varieties without pectin addition, the lowest active acidity (pH) (Figure 2A) after 8 h of cultivation was found in the LB (*L. bulgaricus*) and ST (*S. thermophilus*) varieties, and it is well known that this LAB is one of the most effective acid producers. The titratable acidity of the LB and ST variants was at 62–63 °T, but the ProBio variant had the highest titratable acidity at a higher pH value. Heterofermentative LABs that can form acids characterised by low production of free hydrogen ions are used in ProBio Starter. 

The pH is lowered and the titratable acidity is increased by adding pectin to the oat base. Different beverage variants showed different effects of pectin addition on pH and titratable acidity. The active acid increased in the ST and ProBio varieties and decreased in the Symb and, in particular, the LB varieties. The sum of all acids formed during fermentation in the presence of pectin (Figure 2B) increased significantly in ST and Symb, and decreased in the ProBio variant compared to the pectin-free variant. The titratable acidity of LB sample almost did not change. 

The utilisation of carbohydrates, especially glucose, was closely related to the level of acid production. In the control variants without pectin, glucose from the ST and ProBio starters was most completely utilised, with the lowest concentration and percentage of residual glucose found in these variants (Figure 2C,D). The glucose content of the oat base was increased by the addition of pectin. In the LB and Symb varieties, the addition of pectin resulted in an intensification of glucose uptake, as evidenced by lower total glucose and percentage residual glucose compared to the varieties without pectin. Bacteria in the ST and ProBio varieties became less active in glucose utilisation after pectin addition. The percentage of residual glucose was 20–40% higher than in the corresponding varieties without pectin.

The raw materials of vegetable origin are a source of polyphenolic compounds [46] and these compounds are modified under the action of LAB. After fermentation, the total amount of phenolic compounds detected in the product and in the PFE increased in all varieties (control), with the highest increase observed in the LB variety for product and ST and ProBio in PFE (Figure 3A,B). Pectin is a source of polyphenolic compounds, so it is not surprising that their levels increased in the unfermented oat base with the addition of one. However, no such increase was observed in the fermented ready products, the TPC level in all variants with pectin was approximately at the level of the unfermented control with pectin (red line) (Figure 3A). The TPC of each of the variants tested in the PFE even showed a decrease below the initial level in the unfermented oat base, but above the level of the respective variants without pectin (Figure 3B). 

The water-soluble polysaccharides that can be extracted with alcohol are the sum of the exopolysaccharides synthesised by LAB and the plant polysaccharides derived from the oat raw material. For ease of presentation, we will call them extractable polysaccharides or EPSs. The amount of EPSs after fermentation in variants without pectin increased in all samples (Figure 4). The observed increase in the extractability of EPSs could be due to the synthesis of exopolysaccharides by the lactic acid bacteria [47,48]. The most EPSs were in the Symb sample using multiple cultures in lactic acid bacteria and the least EPSs were in the LB sample using a mono-starter. 

The situation was different for the pectin variants, where as expected the addition of pectin increased the amount of EPSs in the oat base. However, the pectin samples had less EPSs after fermentation than the original oat base (blue line). Firstly, it is an indication of the hydrolysis of pectin under the action of lactic acid bacteria. The lowest amount of EPSs was in the LBs and ProBio samples, it seems that these hydrolysis processes are most pronounced in these samples. The highest amount of EPSs in the variants with pectin was in the Symb sample. Probably the hydrolytic processes of the pectin are compensated by the synthesis of bacterial EPSs in this case.

The protein content of fermented oat base indicates that all samples contained comparable amounts of protein, fat and dry matter regardless of starter type (Table 3). The addition of pectin to the oat base led to an insignificant increase in the measured protein concentration (0.2–0.3%). This is probably due to the increased availability of the measured protein, which can be released from the plant matrix due to the surfactant properties of pectin. The pectin surfactant properties have been described by several authors [49,50]. The experimental samples contained more dry matter, which is due to the presence of pectin. 

#### 3.2.2. Sensory Estimation

Sensory evaluation showed that all samples had a pleasant colour, taste and odour (Figure 5). The hydrolysed non-fermented samples had a much worse sensory profile. The oat base without pectin was evaluated at 3 points (grey solid line). The oat base with pectin was evaluated at 6 points (yellow solid line). Thus, the processing profound changes in the original raw material.

In the control fermented samples, the consistency score was lower, and the respondents had noticeable signs of spread in the drink. In the experimental samples, the consistency was more viscous with the pectin applied, without any separation. Using pectin gave the tasters a pleasant enveloping sensation, resulting in a longer aftertaste. The presence of pectin also added fruitiness to the oats. The Sym_pectin sample had the highest overall organoleptic score (30 points) and this sample’s consistency was the most pleasant, “velvety”.

Fermented milk, including plant-fermented products, is characterised by several specific properties [51,52,53], in particular viscosity, it seemed appropriate to study these parameters in the samples studied. In addition, the results of objective methods (in comparison with sensory evaluation) are preferable for the assurance of quality stability under production conditions, as they have clearer values for control.

#### 3.2.3. Structural and Textural Properties

For fermented milk products and their plant-based analogues, which are heterogeneous systems (emulsions, suspensions), viscosity, water holding capacity (WHC) and tendency to separate (syneresis) are important from the point of view of stability and consumer preference. As expected, the viscosity of all control samples was lower than that of the experimental samples with pectin. The lowest viscosity values in the experimental group were in sample_BP, while the highest values were in sample_TS and sample_Simb, and the viscosity loss (Lη) was reversed (Table 4). There was a natural increase in viscosity loss (Lη) in the experimental samples with respect to the control. 

Syneresis decreased and WHC increased as expected after pectin addition to oat base. Fermentation of the oat base with all starters resulted in improved syneresis and WHC values, regardless of the presence of pectin. The best WHC/syneresis ratio was observed in the ST and Symb samples with pectin. It should be noted that in the Probio_pectin sample, despite the low WHC, the value of syneresis was minimal which indicates the formation of a stable food matrix. The LB sample with pectin had the highest syneresis (Syn = 14.4%) despite high WHC (41%), this ratio indicates that the matrix organisation of the sample is not strong and that it is likely to delaminate.

Other important texture indicators are rheological quality indicators of fermented products. They allow for obtaining objective data on the textural properties of products [54,55,56]. And in conjunction with organoleptic evaluation—to give a more complete picture of the sensory perception of the product. The results of texture profile analysis are presented in Table 5. The hardness index of the experimental samples (+pectin) was slightly higher than the corresponding control samples. These data correlate with the viscosity studies of the test samples, as higher viscosity and thickening of the sample leads to higher hardness. The maximum hardness values were observed in samples ST and Symb with pectin. The highest cohesion and gumminess values were observed in the ProBio and Symb samples with pectin. The Symb_pectin sample showed the highest adhesion. 

The data obtained are of great practical importance, since gumminess and adhesion are reflected in the ability of the product to envelop the oral cavity, creating a full, harmonious and deep flavour, also due to the increased ability to adhere to surfaces. It is undeniable that the addition of pectin leads to the formation of improved structural and viscous properties. In addition to pectin, the changes in the texture of the plant-based drink are also due to the starter used to ferment the vegetable base. 

#### 3.2.4. Antioxidant Properties

The antioxidant properties of the beverages were evaluated using three parameters: Ferric-reducing antioxidant power (FRAP), Radical-scavenging (RSA) and OH free-radical-scavenging ability (OH-SA). The FRAP level in all samples without pectin after fermentation was lower than the initial level (oat_base) (Figure 6A), which may be due to a decrease in the level of glucose, which has reducing properties [57]. The addition of pectin to the oat base increased FRAP, and after fermentation, this activity increased in the ST and ProBio variants and remained at the control level in the LB and Symb samples. 

The LB variant without pectin had a 10% increase in RSA, whereas the other samples maintained activity at the level of the oat base (Figure 6B). The addition of pectins to the oat base did not increase RSA, but after fermentation there was a statistically significant increase in the free radical-scavenging capacity of the pectin-containing beverages. RSA was highest in the LB sample with pectin (60%).

The OH-SA of the oat base was two times lower than that of the RSA. However, the addition of pectin doubled the OH-SA (Figure 6C). After fermentation with LB, ProBio and Symb starter, OH-SA levels increased in samples without pectin. In the samples with pectin, OH-SA increased statistically significantly after fermentation compared to the control only in the ProBio sample. It did not change in the ST and Symb samples, and even decreased significantly in the LB sample (red line). The formation of antioxidant properties is dependent on the additional ingredients used in the beverage and on the starter cultures that are used.

## 4. Discussion

The reason for using oat milk as the base for oat yoghurt is its proven hypocholesterolemia effect, which helps to reduce low-density lipoprotein (LDL) cholesterol levels and is associated with a reduced risk of heart disease. Oats are also high in vitamin B1, which is essential for maintaining energy metabolism. Beta-glucans, a phytochemical found in oat milk, have been studied for their potential health benefits, including lowering cholesterol and improving heart health [42,44]. This milk contains prebiotic fibre, such as fructooligosaccharides, which have been shown to promote the growth of beneficial gut bacteria and improve digestive health [43]. Oat milk contains higher levels of vitamins and minerals, such as vitamin E, magnesium and potassium, than cow’s milk. This suggests that plant-based dairy alternatives can provide the body with valuable nutrients that may not be present in milk [58]. Accordingly, the introduction of beneficial bacteria into oat milk and its subsequent fermentation will only enhance its therapeutic and preventive properties. 

A comprehensive data analysis showed that the use of pectin in combination with oat base significantly affected the position of oat base and oat base + pectin points on a coordinate plane (Figure 7A). Fermented oat drinks form two domains depending on the presence of pectin; these domains do not overlap, indicating a significant difference in their sum properties. Comparing the properties of the drinks with only fermented oat (Figure 7B), it was found that the beverages without pectin were similar in their properties and formed a single group, the LB sample was similar to ProBio, and the ST samples were similar to Symb. The sum of the properties of the LB_pectin sample became more characteristic than the other ProBio, ST and Symb beverages when pectin was used.

These starter cultures were selected because of their widespread use in fermented milk products and their commercial availability. Here, our interest was to test the possibility of using dairy starter cultures for the manufacture of products based on vegetable raw materials. The analysis of the results also showed that the strains used in starter cultures have a positive effect on the health-promoting properties of the oat beverages. Undoubtedly, the characteristics of the starter cultures and the different metabolic properties of the LABs used in their composition are responsible for the differences in the changes in antioxidant activities after fermentation. Furthermore, pectin undergoes metabolic transformations depending on the starter cultures and plays an important role in the development of antioxidant properties. Between the level of RSA (DPPH) and the total amount of phenolic compounds in the product, a high positive correlation was found (Figure 7C). OH-SA is correlated with extractable polysaccharide (EPS) content, whereas it is less correlated with phenolic compounds. In plant raw materials, phenolic compounds can be present in a variety of states, in particular in free and bound forms. Phenolic compounds can be bound to cell wall components (cellulose, hemicellulose, lignin and pectin) [59]. Li et al., 2023 showed that the fermentation of oat bran leads to an increase in the content of phenolic compounds, including their release from cellular structures [60]. Undoubtedly, the characteristics of the starter cultures and the different metabolic properties of the LABs used in their composition are responsible for the differences in the changes in antioxidant activities after fermentation. Furthermore, pectin undergoes metabolic transformations.

Certainly, the differences we found are due to the type of fermentation starter used. The usage of *L. bulgaricus* (LB) mono-starter allows for efficient glucose utilisation, as evidenced by a higher accumulation of lactic acid with a lower glucose conversion. The addition of pectin improves glucose conversion (lower percentage of residual glucose in the product) without increasing lactic acid synthesis. The *L. bulgaricus* drink is characterised by high adhesion and syneresis and low cohesion compared to the other drinks. Although this beverage has high antioxidant activity, it has a low sensory profile. Fermentation with *L. bulgaricus*, *S. thermophilus* and *Lactococcus* (Symb) in an oat-based beverage produced a product with high appetite, rich body flavour and low gel strength. A particular attribute of this starter is the presence of *L. lactis* ssp. *cremoris*, *L. lactis* ssp. *lactis*, *L. lactis* ssp. *lactis var. diacetylactis*. It is probably the lactococci that are responsible for the formation of a viscous matrix in the beverage, characterised by a low syneresis, a high WHC, a gummy texture and adhesiveness. The Symb sample had the highest EPS content, possibly due to the presence of lactococci, as many researchers have reported their high EPS synthesising activity [61,62,63]. On the other hand, the presence of *L. casei*, *L. rhamnosus* and *L. paracasei* strains in the ProBio starter yields a product with high antioxidant activity, especially in the presence of pectin. Indeed, the high antioxidant potential of these LAB species has been written about by many researchers [64,65,66,67].

Oats are an excellent source of phenolic compounds and have antioxidant activity [68,69]. They are also a prebiotic, helping to preserve the probiotic bacteria included in Symb and ProBio starter, and the use of pectin helps to increase the antioxidant properties, also due to its phenolic compounds [70,71]. 

It is necessary to create a texture that is acceptable to the consumer in order to produce a beverage that is not only healthy but also tasty. One of the most useful ingredients in this regard is pectin. Its successful use in creating the texture of plant-based non-dairy beverages has been described by many [72,73,74,75]. In agreement with our results, the use of pectin has a positive effect on the textural and sensory characteristics of plant-based beverages and that it reduces the syneresis. As a result, this comprehensive study provides valuable information on the effects of specific starter/probiotic combinations on the textural, rheological, antioxidant and sensory properties of oat-based yogurts. The results show that it is possible to select the most suitable starter cultures to meet consumer preferences and beverage specifications.

The potential commercial use of these plant-based beverages is very diverse. For example, a complete replacement of livestock products. This is important for ethical, religious and other reasons. Providing full nutrition to people who voluntarily gave up meat (vegans, vegetarians, etc.) Expanding the range of fermented products containing prebiotics and probiotics.

## 5. Conclusions

Theoretical substantiation and practical confirmation prove the viability of using oats to create alternative fermented products from plant raw materials. This makes it possible to offer the grain drink as a matrix for a number of new food products with a complex of healthy properties. The study highlights the potential for the use of multi-component starter cultures of lactic acid bacteria that, due to their exopolysaccharide synthesis properties, are better able to form the structure of a fermented vegetable beverage. Adding pectin to oat drinks significantly improves the viscosity, texture and antioxidant properties, which has a positive effect on the sensory evaluation of the product and increases its usefulness. The results also showed that the formation of the complex of drink properties depends on the type of starter and the presence of pectins. Understanding the formation of the matrix of the ‘vegetable’ clot during fermentation, as well as its behaviour during maturation and storage, will be the subject of further research. We will also investigate the most suitable way to use the fermented oat base to make finished products like dairy products, sauces, dressings, desserts and others.

## Figures and Tables

**Figure 1 biotech-13-00038-f001:**
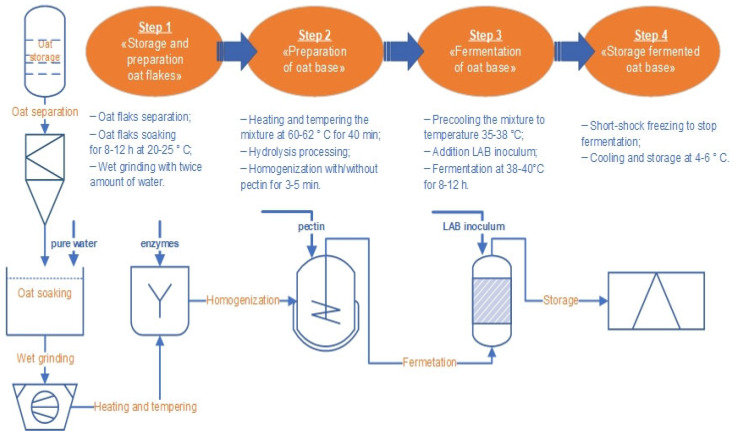
The production steps of the fermented oat base.

**Figure 2 biotech-13-00038-f002:**
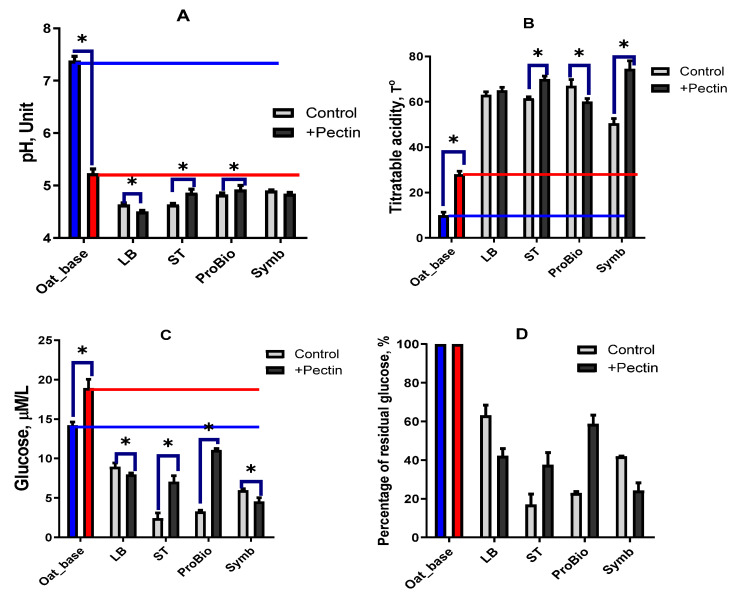
Effect of starters and pectin on pH (**A**) and titratable acidity (**B**), absolute glucose concentration (**C**) and percentage of residual glucose (**D**). Blue bar—unfermented oat base without pectin, red bar—unfermented oat base with pectin, grey bar—oat-based fermented beverage without pectin, black bar—oat-based fermented beverage with pectin. The blue line shows the level of the index for the oat base without pectin and the red line shows the level of the index for the oat base with pectin. Asterisks indicate statistically significant differences between variants without pectin (control) and with pectin according to non-parametric one-way analysis of variance (Kruskal–Wallis) test, *p* < 0.05.

**Figure 3 biotech-13-00038-f003:**
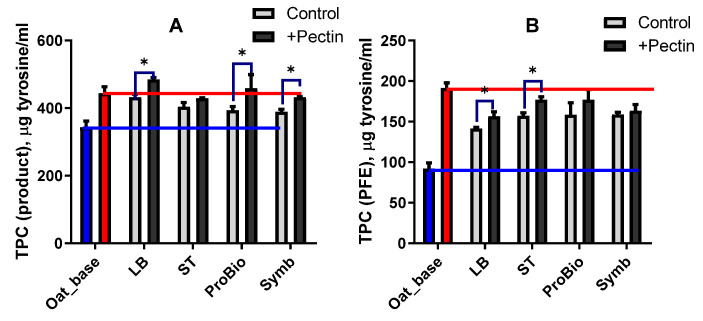
Effect of starter and the pectin to concentration on total phenol-containing compounds (TPCs) ((**A**), product; (**B**), protein-free extract). Blue bar—unfermented oat base without pectin; red bar—unfermented oat base with pectin; grey bar—oat-based fermented beverage without pectin; black bar—oat-based fermented beverage with pectin. The blue line shows the level of the index for the oat base without pectin and the red line shows the level of the index for the oat base with pectin. Asterisks indicate statistically significant differences between variants without pectin (control) and with pectin according to non-parametric one-way analysis of variance (Kruskal–Wallis) test, *p* < 0.05.

**Figure 4 biotech-13-00038-f004:**
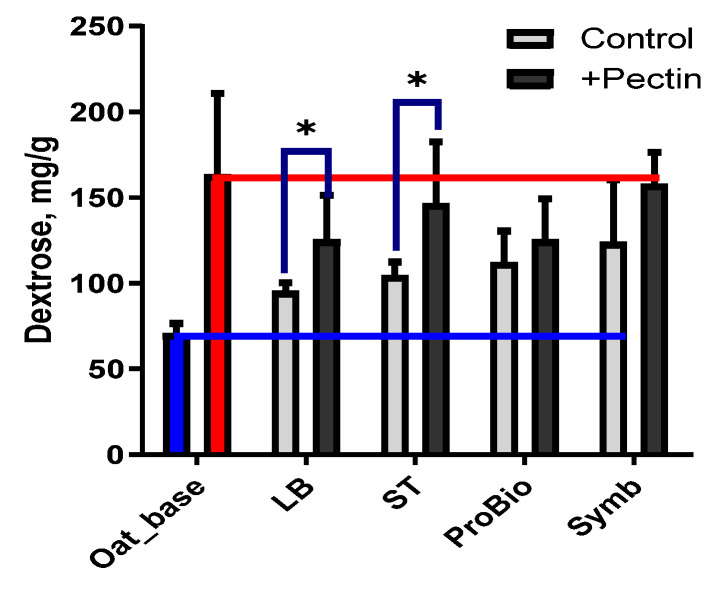
Total amount of extractable polysaccharides (EPSs). Blue bar—unfermented oat base without pectin; red bar—unfermented oat base with pectin; grey bar—oat-based fermented beverage without pectin; black bar—oat-based fermented beverage with pectin. The blue line shows the level of the index for the oat base without pectin and the red line shows the level of the index for the oat base with pectin. Asterisks indicate statistically significant differences between variants without pectin (control) and with pectin according to non-parametric one-way analysis of variance (Kruskal–Wallis) test, *p* < 0.05.

**Figure 5 biotech-13-00038-f005:**
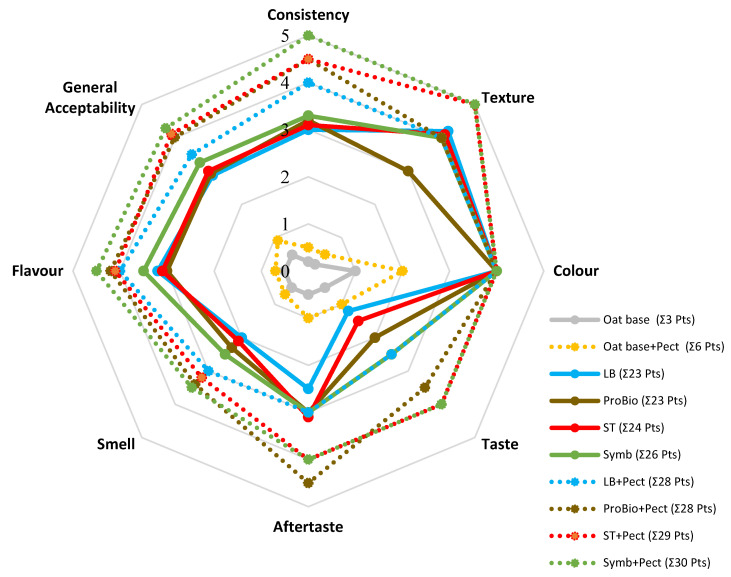
Sensory profile of oat-based fermented beverage. Controls without pectin (LB, ST, ProBio, Symb and oat base) are indicated by the thick solid lines, samples with pectin (LB + pectin, ST + pectin, ProBio + pectin, Symb + pectin and oat base + pectin) are indicated by the dashed lines.

**Figure 6 biotech-13-00038-f006:**
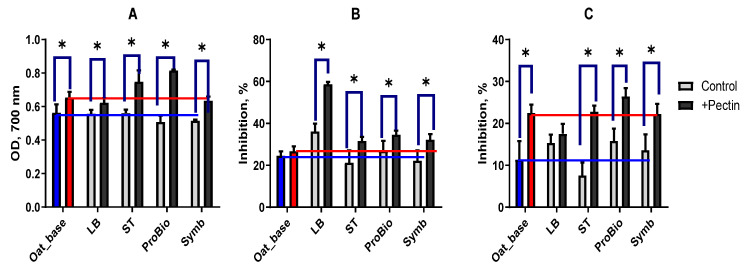
Antioxidant activity of fermented oat base: Ferric-reducing antioxidant power (**A**); Radical-scavenging ability (**B**); OH free-radical-scavenging ability (**C**). Blue bar—unfermented oat base without pectin, red bar—unfermented oat base with pectin, grey bar—oat-based fermented beverage without pectin, black bar—oat-based fermented beverage with pectin. The blue line shows the level of the index for the oat base without pectin and the red line shows the level of the index for the oat base with pectin. Asterisks indicate statistically significant differences between variants without pectin (control) and with pectin according to non-parametric one-way analysis of variance (Kruskal–Wallis) test, *p* < 0.05.

**Figure 7 biotech-13-00038-f007:**
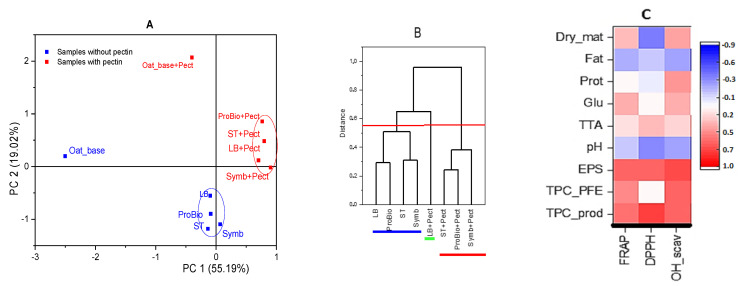
Principal component analysis (**A**) of oat base and oat-based fermented beverages with/without pectin, dendrogram of oat-based fermented beverages with/without pectin (**B**) and heat-map correlation (**C**) of antioxidant activity and chemical composition.

**Table 1 biotech-13-00038-t001:** The description of samples.

Sampel	Starter Cultures			Short Description
	*Lactobacilli*	*Strepto*/*Lactococci*	*Bifidobacteria*	
**LB**	*L. delbrueckii subsp. bulgaricus*	-	-	The oat base was treated by amylolytic enzymes and then fermented by appropriate LAB starter cultures.
**ST**	-	*S. thermophilus*	*-*	
**ProBio**	*L. delbrueckii* ssp. *bulgaricus* *L. acidophilus*, *L. casei*, *L. rhamnosus*, *L. paracasei*	*S. thermophilus*	*B. lactis*, *B. infantis*	
**Symb**	*L. delbrueckii* ssp. *bulgaricus* *L. acidophilus*	*S. thermophilus* *L. lactis* ssp. *Cremoris*, *L. lactis* ssp. *lactis*, *L. lactis* ssp. *lactis var. diacetylactis*	*B. lactis*	
**Control**	Oat_base without pectin was fermented by starter cultures LAB			
**+Pectin**	Oat_base with pectin (1%) was fermented by starter cultures LAB			

**Table 2 biotech-13-00038-t002:** Parameters of the oat base before and after hydrolysis (mean values ± SD, n = 3). Asterisks indicate statistically significant differences between variants before and after hydrolysis according to non-parametric one-way analysis of variance (Kruskal–Wallis) test, *p* < 0.05.

Parameter	Before	After
pH	7.54 ± 0.08 *	7.34 ± 0.07 *
Titrable acidity, °T	8.0 ± 0.05 *	10.0 ± 0.06 *
Protein, %	2.66 ± 0.08 *	3.07 ± 0.25 *
Fat, %	1.33 ± 0.07 *	1.66 ± 0.21 *
Dry matter, %	12.37 ± 0.09 *	14.80 ± 0.38 *
Total sugar, %	2.73 ± 0.02 *	6.36 ± 0.05 *
Glucose, mmol/L	0.1 ± 0.01 *	16.0 ± 0.02 *

**Table 3 biotech-13-00038-t003:** Chemical parameters of oat-based fermented beverage.

Sample	Protein, %		Fat, %		Dry Matter, %	
	Control	+Pectin	Control	+Pectin	Control	+Pectin
**Oat_base**	3.07 ± 0.25	3.29 ± 0.15 ^b^	1.33 ± 0.03	0.86 ± 0.01 ^b^	11.81 ± 0.32	13.56 ± 0.24 ^b^
**LB**	3.23 ± 0.06	3.24 ± 0.06	0.91 ± 0.02 ^a^	0.94 ± 0.02 ^a^	10.74 ± 0.21 ^a^	11.10 ± 0.22 ^a^
**ST**	3.10 ± 0.06	3.40 ± 0.07 ^b^	0.81 ± 0.02 ^a^	0.94 ± 0.02 ^ab^	11.39 ± 0.23 ^a^	11.99 ± 0.24 ^ab^
**ProBio**	3.47 ± 0.17 ^a^	3.44 ± 0.12	0.97 ± 0.02 ^a^	0.91 ± 0.02 ^ab^	11.48 ± 0.23 ^a^	12.09 ± 0.24 ^ab^
**Symb**	3.25 ± 0.07	3.16 ± 0.06 ^a^	0.92 ± 0.02 ^a^	0.89 ± 0.02	11.12 ± 0.22 ^a^	11.58 ± 0.23 ^ab^

^a^ indicates statistically significant differences between variants without pectin (control) and with pectin according to non-parametric one-way analysis of variance (Kruskal–Wallis) test, *p* < 0.05. ^b^ indicate statistically significant differences between oat_base (oat_base + pectin) and fermented beverages to non-parametric one-way analysis of variance (Kruskal–Wallis) test, *p* < 0.05.

**Table 4 biotech-13-00038-t004:** Viscosity, structure and thixotropic properties of oat-based fermented beverage.

Sample	Apparent Viscosity, mPa·s^−1^		L_η_, %		CMR	
	−Pectin	+Pectin	−Pectin	+Pectin	−Pectin	+Pectin
**Oat_base**	22.4 ± 1.1	40.3 ± 2.0 ^ab^	6.5	3.5	0.87	0.96
**LB**	26.3 ± 1.3 ^a^	44.6 ± 2.2 ^ab^	4.2	2.4	0.94	0.98
**ST**	68.8 ± 3.4 ^a^	146.8 ± 7.3 ^ab^	4.4	2.0	0.94	0.98
**ProBio**	59.5 ± 3.0 ^a^	101.2 ± 5.1 ^ab^	5.2	2.5	0.92	0.97
**Symb**	64.8 ± 3.2 ^a^	141.7 ± 7.1 ^ab^	3.0	1.0	0.94	0.98
**Sample**	**Syn, %**		**Δ Syn, % to −Pectin**	**WHC, %**		**Δ WHC, % to −Pectin**
	**−Pectin**	**+Pectin**		**−Pectin**	**+Pectin**	
**Oat_base**	45.0 ± 2.3 ^ab^	28.0 ± 1.4 ^ab^	−42.6	23.0 ± 1.2 ^ab^	32.0 ± 1.6 ^ab^	+35.4
**LB**	36.9 ± 1.8 ^ab^	14.4 ± 0.7 ^ab^	−61.1	27.6 ± 1.4 ^ab^	41.0 ± 2.1 ^ab^	+48.7
**ST**	27.4 ± 1.4 ^ab^	3.5 ± 0.2 ^ab^	−87.2	28.0 ± 1.4 ^ab^	39.0 ± 2.0 ^ab^	+39.3
**ProBio**	31.0 ± 1.6 ^ab^	3.7 ± 0.2 ^ab^	−81.0	29.3 ± 1.5 ^ab^	32.5 ± 1.6 ^ab^	+11.0
**Symb**	27.5 ± 1.4 ^ab^	5.2 ± 0.3 ^ab^	−88.1	29.9 ± 1.5 ^ab^	41.5 ± 2.1 ^ab^	+38.9

^a^ indicates statistically significant differences between variants without pectin (control) and with pectin according to non-parametric one-way analysis of variance (Kruskal–Wallis) test, *p* < 0.05. ^b^—indicate statistically significant differences between oat_base (oat_base + pectin) and fermented beverages to non-parametric one-way analysis of variance (Kruskal–Wallis) test, *p* < 0.05.

**Table 5 biotech-13-00038-t005:** The texture parameters of samples.

Sample	Hardness, g		Cohesiveness, %	
	−Pectin	+Pectin	−Pectin	+Pectin
**LB**	11.22 ± 0.56	12.11 ± 0.61	94.34 ± 4.72	97.61 ± 4.88
**ST**	12.53 ± 0.63	13.21 ± 0.69	75.90 ± 3.80	81.51 ± 4.08
**ProBio**	11.92 ± 0.60	13.12 ± 0.66 ^a^	92.68 ± 4.63	96.37 ± 4.92
**Symb**	11.93 ± 0.60	12.91 ± 0.65	92.19 ± 4.61	95.84 ± 4.79
**Sample**	**Gumminess, g**		**Adhesiveness, g·mm**	
	**−Pectin**	**+Pectin**	**−Pectin**	**+Pectin**
**LB**	10.60 ± 0.53	11.81 ± 0.59 ^a^	45.44 ± 2.27	51.53 ± 2.58 ^a^
**ST**	9.5 ± 0.48 ^b^	11.16 ± 0.56 ^a^	52.2 ± 2.61 ^b^	56.85 ± 2.84 ^a^
**ProBio**	11.72 ± 0.59 ^b^	12.12 ± 0.61 ^a^	49.63 ± 2.48 ^b^	54.86 ± 2.74 ^a^
**Symb**	10.97 ± 0.55	12.38 ± 0.62 ^a^	51.28 ± 2.56 ^b^	60.22 ± 3.01 ^ab^

^a^—indicate statistically significant differences between variants without pectin (control) and with pectin according to non-parametric one-way analysis of variance (Kruskal–Wallis) test, *p* < 0.05. ^b^—indicate statistically significant differences between oat_base (oat_base + pectin) and fermented beverages to non-parametric one-way analysis of variance (Kruskal–Wallis) test, *p* < 0.05.

## Data Availability

The original contributions presented in the study are included in the article/Appendix A; further inquiries can be directed to the corresponding author.

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
