# Peer review of "Chemical, Textural and Antioxidant Properties of Oat-Fermented Beverages with Different Starter Lactic Acid Bacteria and Pectin"

_biotech, 2024, doi:10.3390/biotech13040038_

Round 1

Reviewer 1 Report

Comments and Suggestions for Authors

Introduction : The introduction provides a good overview of the oat-based fermented beverages and the role of lactic acid bacteria, also pectin substances.

Materail and methods

1. Specify the source and type of pectin used in the study (e.g., high-methoxyl, low-methoxyl) and the concentrations used in the different formulations.

2. It would be beneficial to include the conditions under which enzyme treatment was carried out (e.g., temperature, time). This will aid in replicability.

2. Check the reference in each experiment (TPC, FARP….)

Results

·       3.1 the experimental methods should be moved to Material and method section.

·       Compare your findings with other studies on fermented oat beverages or similar fermented foods.

·       Table 2 add statistical analysis to compare the differences data

·       Figure 6 for FARP assay, provide unit of antioxidant activity

Discussions

What are the potential commercial applications of these oat-fermented beverages? How do these findings contribute to the development of functional foods?

Ethical Considerations:

Because human sensory evaluation was part of the study, mention whether ethical approval was obtained. Given that oats are a potential allergen, it is crucial to address how this risk was managed.

References

Please ensure that all references are up-to-date and correctly formatted according to the journal's guidelines.

Author Response

Dear reviewer, thank you for your comments, they helped to improve our article. The responses to your comments are attached. Changes in the file are highlighted in yellow.

Com1 Introduction : The introduction provides a good overview of the oat-based fermented beverages and the role of lactic acid bacteria, also pectin substances.

Res1 Thank you, we tried to write a good introduction

Com2 Material and methods 1. Specify the source and type of pectin used in the study (e.g., high-methoxyl, low-methoxyl) and the concentrations used in the different formulations.

Res2 “Pectin (commercial apple high methoxyl pectin ("Pectowin", Poland) in an amount of 1% was added to half of the samples.” was added in paragraph 2.2

Com3 It would be beneficial to include the conditions under which enzyme treatment was carried out (e.g., temperature, time). This will aid in replicability.

Res3 The mixture was fermented at a temperature of 60 ± 2 ° C for 40 minutes. This is a very important process. Therefore, we have already described this in the article (paragraph 2.1).

Com4 Check the reference in each experiment (TPC, FARP….)

Res 4 We have checked all the references again as you requested. They are all correct.

Com 5  Results 3.1 the experimental methods should be moved to Material and method section.

Res 5 Dear reviewer, let us disagree and not move part 3.1. to ‘materials and methods’. This is especially important, due to the fact that these technological methods and results will be the basis for writing subsequent articles, and will be used by us for citations. Presenting these materials in the ‘results’ part will allow us to cite the preparation of oat raw material for fermentation more clearly.

Com 6 Compare your findings with other studies on fermented oat beverages or similar fermented foods

Res6 We reviewed similar studies in this field: Kumar, L. et. al., 2021; Charalampopoulos, D. et. al., 2002; Wronkowska, M. et. al., 2022; Dallagnol, A.M. et. al., 2013; Chen, L. et. al., 2020. Many experts consider herbal beverages to be very promising and beneficial. One of the main problems is the poor texture of the product. We studied the effect of lactic acid bacteria on texture. And also the possibility of texture correction by pectin was studied.

Com 7 Table 2 add statistical analysis to compare the differences data

Res 7 we added Parameters of the oat base before and after hydrolysis (mean values ± SD, n = 3) . Asterisks indicate statistically significant differences between variants before and after hydrolysis according to non-parametric one-way analysis of variance (Kruskal-Wallis) test, p < 0.05.

Com 8 Figure 6 for FARP assay, provide unit of antioxidant activity

Res 8 Thank you, corrected

Com9 Discussions. What are the potential commercial applications of these oat-fermented beverages? How do these findings contribute to the development of functional foods?

Res9  Added. The potential commercial use of these plant-based beverages is very diverse. For ex-ample, a complete replacement of livestock products. This is important for ethical, religious and other reasons. Providing full nutrition to people who voluntarily gave up meat (vegans, vegetarians, etc.) Expanding the range of fermented products containing prebiotics and probiotics. (line 477-481)

Com10  Ethical Considerations: Because human sensory evaluation was part of the study, mention whether ethical approval was obtained. Given that oats are a potential allergen, it is crucial to address how this risk was managed.

Res10 “Subjects were asked in advance if they had any food allergies, as oats have the potential to cause it. Ethical approval was also obtained for the publication of the results“ were added in paragraph 2.10.

Com11 References Please ensure that all references are up-to-date and correctly formatted according to the journal's guidelines.

Res 11 We have checked all the references again as you requested. They are all correctly formatted according to the journal's guidelines. 

Reviewer 2 Report

Comments and Suggestions for Authors

1. How to get protein-free extract? Why study the effect of the starter and the pectin concentration on the TPC of protein-free extract? 

 2. Total phenolic compounds (TPC) assays: Do glucose and pH significantly affect results and cause errors?

3. Why was the protein content of the oat base without pectin lower than that of the oat base with pectin?

4. Figure 5. Sensory profile of oat-based fermented beverage: Why not show the results of the controls?

5. The FRAP level in all samples without pectin after fermentation was lower than the initial level, which may be due to a decrease in the level of glucose, which has reducing properties: Why is there no correlation between FRAP values and glucose content in different samples?

6. The analysis of the content of protein and fat in the samples were determined by near infrared spectroscopy. Some increase in protein and fat is probably due to their release from the native protein-fat-carbohydrate matrix under the action of an enzyme and heating (line 259-261): Does the release affect the protein and fat contents measured by NIR?

 7. Line 253-255: To prepare the oat base, the suspension obtained previously was hydrolysed with an amylolytic agent at 60 ± 2°C for 40 minutes, monitoring the increase in glucose content. At the end of the operation, the mixture was heated to 95-98 °C for 5 minutes to inactivate enzymes and increase the stability of the food system: This information should be included in the Materials and Methods section.

 8. There is insufficient discussion or introduction of why you chose these combinations of starter strains and how the strain's metabolic properties  affect the nature of the product.

Comments on the Quality of English Language

Line 137-138: Five ml of 1% trichloroacetic acid solution was added to five g of sample and mixed [37].

Line 147-148: Isolation and quantification of extractable polysaccharides were carried out as described by [37].

Line 262: The «flour» flavor

Line 329: The lowest amount of EPS was found in the LBs and ProBio samples)

Line 379: water loss (Syn=37.9%)

min or minutes

Author Response

Dear reviewer, thank you for your comments, they helped to improve our article. The responses to your comments are attached. Changes in the file are highlighted in yellow.

Com 1 How to get protein-free extract? Why study the effect of the starter and the pectin concentration on the TPC of protein-free extract? 

Res 1 The preparation of protein-free extract is described in lines 137-139, paragraph 2.4.

Pectin is a source of phenolic compounds, they can be in bound and free form. The analysis of TPCs in PFE allows us to identify precisely the low molecular weight TPCs, which are more likely to exhibit antioxidant properties. Several studies have shown that the total phenolic compound content can be used as an indicator of antioxidant activity, although the total phenolic compound content does not include all antioxidants.

In addition, our use of different starter leads to differences in metabolites, low molecular weight phenol-containing compounds are released from the plant substrate, accordingly, we were interested in these differences as they affect the antioxidant properties of plant-based yoghurt. 

Com2 Total phenolic compounds (TPC) assays: Do glucose and pH significantly affect results and cause errors?

Res2 We found no correlations with pH, glucose level and TPC amount. Taking into account the previous experience of other studies and our own experience, the specificity of the using Folin–Ciocalteu reaction, the use of this method for TPC analysis is reasonable and there is no significant distortion of the results under the influence of glucose and pH. 

Com 3 Why was the protein content of the oat base without pectin lower than that of the oat base with pectin?

Res 3 The protein content of the oat base is higher by 0.24 % due to the increased density of the beverage as a result of the introduction of pectin, here the volume-density relationship plays a role.

Com 4 Figure 5. Sensory profile of oat-based fermented beverage: Why not show the results of the controls?

Res 4 Corrected and added. Figure 5. Sensory profile of oat-based fermented beverage. Controls without pectin (LB, ST, ProBio, Symb) are indicated by the thick solid lines, samples with pectin (LB+pectin, ST+pectin, ProBio+pectin, Symb+pectin) are indicated by the dashed lines.

Com 5 The FRAP level in all samples without pectin after fermentation was lower than the initial level, which may be due to a decrease in the level of glucose, which has reducing properties: Why is there no correlation between FRAP values and glucose content in different samples?

Res 5 Thank you for your question, you are quite right, in our other research on milk yoghurt we have observed a direct correlation between glucose levels and FRAP. In our studies on plant-based fermented beverages, we found a low correlation, which seems to be due to the presence of other compounds with restorative properties: polyphenols, plant pigments, and others.

Com 6 The analysis of the content of protein and fat in the samples were determined by near infrared spectroscopy. Some increase in protein and fat is probably due to their release from the native protein-fat-carbohydrate matrix under the action of an enzyme and heating (line 259-261): Does the release affect the protein and fat contents measured by NIR?

Res 6 Based on the results of our present research, previous studies, the peculiarity of plant substrate is the high binding of food components (proteins, fats) in the carbohydrate matrix, so during the fermentation process we observe an increase in the level of proteins and fats in the product detected by NIR

Com 7 Line 253-255: To prepare the oat base, the suspension obtained previously was hydrolysed with an amylolytic agent at 60 ± 2°C for 40 minutes, monitoring the increase in glucose content. At the end of the operation, the mixture was heated to 95-98 °C for 5 minutes to inactivate enzymes and increase the stability of the food system: This information should be included in the Materials and Methods section.

Res 7  Thank you, corrected (#2.1)

Com 8 There is insufficient discussion or introduction of why you chose these combinations of starter strains and how the strain's metabolic properties  affect the nature of the product.

Res 8 Added: These starter cultures have been selected because of their widespread use in fermented milk products and their commercial availability. Here, our interest was to test the possibility of using dairy starter cultures for the manufacture of products based on vegetable raw materials. The analysis of the results also showed that the strains used in starter cultures have a positive effect on the health-promoting properties of the oat beverages.

Comments on the Quality of English Language

Line 137-138: Five ml of 1% trichloroacetic acid solution was added to five g of sample and mixed [37]. – corrected

Line 147-148: Isolation and quantification of extractable polysaccharides were carried out as described by [37]. – corrected

Line 262: The «flour» flavor - corrected

Line 329: The lowest amount of EPS was found in the LBs and ProBio samples) - corrected

Line 379: water loss (Syn=37.9%)  - corrected

min or minutes – corrected (min)

Reviewer 3 Report

Comments and Suggestions for Authors

I would like to appreciate the author (s) submitted work. The entire manuscript sounds good to read, got sufficient data to support. A very tiny typo error paragraph 341 as pro-tein. 

Author Response

Dear reviewer! Thank you for reviewing our paper, we appreciate your opinion.

Ans1: tiny typo error paragraph 341 as pro-tein.  

Comments 1: corrected

Round 2

Reviewer 2 Report

Comments and Suggestions for Authors

1.     The preparation of the PFE and the TPC assay should be described separately. Is there any better method for oat, a protein-containing material, than the Folin–Ciocalteu reagent method?

2.     Give a reference or description of sensory evaluation criteria.

      After hydrolysis, the sensory features of the oat base improved significantly. Therefore, the sensory profile of the oat base should be added, the same as other results.

3.     Some increase in protein and fat is probably due to their release from the native protein-fat-carbohydrate matrix under the action of an enzyme and heating: Give a reference.

4.     The observed increase in EP could be due, firstly, to the synthesis of exopolysaccharides by lactic acid bacteria and, secondly, to the increased extractivity of plant polysaccharides resulting from the action of LABs. Give a reference.

5.   The protein content of the oat base without pectin was lower than that of the oat base with pectin, please check this data.

6.     The FRAP level in all samples without pectin after fermentation was lower than the initial level, which may be due to a decrease in glucose level, which has reducing properties: Give a reference.

Comments on the Quality of English Language

In view of the above the aim of this study was to determine the physico-chemical, rheological, antioxidant and sensory properties of oats fermented beverages produced using four different starter cultures widely available on the Russian market, but designed for milk fermentation.

Lines 99-100: The mixture is consisting of crushed oat flakes

Titrable acidity, pH and glucose content were performed as described previously

Line 283: The observed increase in EP could be due

The ferric reducing antioxidant power (FRAP) analysis was performed according to the method described in [41] with modifications.

Line 343: (Syn=14,4%)

Author Response

Dear Reviewer, Thank you for your careful reading of the article, we have taken your comments into account and made the necessary changes and references. 

Com1 The preparation of the PFE and the TPC assay should be described separately. Is there any better method for oat, a protein-containing material, than the Folin–Ciocalteu reagent method?

Ans1  The preparation of the PFE and the TPC assay should be described separately. - Added # 2.4.1 Preparing of protein-free extract (PFE).

There are many protein analysis methods such as Bradford protein assay, bicinchoninic acid assay (BCA assay), and others, however, in our studies we use Folin-Ciocalteu reagent method to detect total phenolic compounds.

It is known that oat bran contains many functional components and phytochemistry substances (Li, Y., Qin, C., Dong, L. Z., Zhang, X., Wu, Z. F., Liu, L. Y., … Liu, L. L. (2022). Whole grain benefit: Synergistic effect of oat phenolic compounds and beta-glucan on hyperlipidemia via gut microbiota in high-fat-diet mice. Food & Function, 13(24), 12686–12696. https://doi.org/10.1039/d2fo01746f). The abundance of phenolic compounds in bran (mainly including ferulic acid, chlorogenic acid, p-hydroxybenzoic acid, rutin, apigenin, vanillin, and luteolin) makes it one of the most crucial resources for natural antioxidants. The phenolic compounds existed mainly in the free and bound states, where the bound phenolic compounds were connected with cell wall components (Li Y, Zhang Y, Dong L, Li Y, Liu Y, Liu Y, Liu L, Liu L. Fermentation of Lactobacillus fermentum NB02 with feruloyl esterase production increases the phenolic compounds content and antioxidant properties of oat bran. Food Chem. 2024 Mar 30;437(Pt 1):137834. doi: 10.1016/j.foodchem.2023.137834. Epub 2023 Oct 21. PMID: 37897817.) 

Com2 Give a reference or description of sensory evaluation criteria.

After hydrolysis, the sensory features of the oat base improved significantly. Therefore, the sensory profile of the oat base should be added, the same as other results.

Ans2 We described of sensory evaluation criteria in Supplementary file.

So, in paragraph 3.3.2, text was added: “The hydrolyzed non-fermented samples had a much worse sensory profile. The oat base without pectin was evaluated at 3 points (gray solid line). The oat base with pectin was evaluated at 6 points (yellow solid line). Thus, the processing profound changes in the original raw material.”

Fig. 5 had also been changed.

Com3 Some increase in protein and fat is probably due to their release from the native protein-fat-carbohydrate matrix under the action of an enzyme and heating: Give a reference.

Ans3 Added: “The standard process for obtaining protein from oats is heating with enzymatic treatment, which significantly increases the yield of protein components from plant raw materials. [Sargautis, D.; Kince, T.; Gramatina, I. Characterisation of the Enzymatically Extracted Oat Protein Concentrate after Defatting and Its Applicability for Wet Extrusion. Foods 2023, 12, 2333. https://doi.org/10.3390/foods12122333]. Due to the short duration of treatment, our case shows a similar effect at a lower level.”

Com4 The observed increase in EP could be due, firstly, to the synthesis of exopolysaccharides by lactic acid bacteria and, secondly, to the increased extractivity of plant polysaccharides resulting from the action of LABs. Give a reference.

Ans4  We changed sentence and added a reference. “The observed increase in EPS could be due to the synthesis of exopolysaccharides by lactic acid bacteria [Mårtensson et al., 2003; Wang et al., 2024]”.

Mårtensson, O., Dueñas-Chasco, M., Irastorza, A., Öste, R., & Holst, O. (2003). Comparison of growth characteristics and exopolysaccharide formation of two lactic acid bacteria strains, Pediococcus damnosus 2.6 and Lactobacillus brevis G-77, in an oat-based, nondairy medium. LWT - Food Science and Technology, 36(3), 353–357.

Y. Wang, S. Mehmood, N. H. Maina, K. Katina, R. Coda. Synthesis in situ of heteropolysaccharide by Levilactobacillus brevis AM7 during fermentation of oat and hemp and its effect on the techno-functional properties of oat yogurt type model. Food Hydrocolloids, Volume 147, Part B, 2024,109416

Com5 The protein content of the oat base without pectin was lower than that of the oat base with pectin, please check this data.

Ans5 Thank you for reading the article carefully, we have taken additional measurements, calculated the results again and corrected Table 3. (highlighted in red)

Com6 The FRAP level in all samples without pectin after fermentation was lower than the initial level, which may be due to a decrease in glucose level, which has reducing properties: Give a reference.

Ans6 The FRAP level in all samples without pectin after fermentation was lower than the initial level, which may be due to a decrease in glucose level, which has reducing properties [Shendurse A.M., and Khedkar C.D. (2016) Glucose: Properties and Analysis. In: Caballero, B., Finglas, P., and Toldrá, F. (eds.) The Encyclopedia of Food and Health vol. 3, pp. 239-247. Oxford: Academic Press].

Comments on the Quality of English Language

Corrected - In view of the above the aim of this study was to determine the physico-chemical, rheological, antioxidant and sensory properties of oats fermented beverages produced using four different starter cultures widely available on the Russian market, but designed for milk fermentation.

Corrected -Lines 99-100: The mixture is consisting of crushed oat flakes

Corrected -Titrable acidity, pH and glucose content were performed as described previously

This sentence has been amended - Line 283: The observed increase in EP could be due

Corrected -The ferric reducing antioxidant power (FRAP) analysis was performed according to the method described in [41] with modifications.

Corrected -Line 343: (Syn=14,4%)

Round 3

Reviewer 2 Report

Comments and Suggestions for Authors

Table 3: Normally pectin contains no protein. The protein content of the oat base without pectin was lower than that of the oat base with pectin, why?

Comments on the Quality of English Language

The writing of the manuscript should be polished more carefully.

Line 131: The analysis of the content of protein, fat, solids and total sugars in the samples were determined by near infrared spectroscopy: was

Line 187: step

Line 198: OH-free radical scavenging ability was carried out following the procedure described by Sungatullina et al. [41].

Line 209: Oat-based products

Line 337: As fermented milk, including plant fermented products, are characterized by a number of specific properties: Fermented milk, including plant-fermented products, is characterized by several specific properties

Lines 422-423: Oat milk contains higher levels of vitamins and minerals, such as vitamin E, magnesium and potassium, than cow's milk.

Line 428: showed that the use of pectin

Line 476: On the other hand

Line 487: In agreement with our results, the use

Author Response

Thank you very much for your careful review of our article, all corrections are highlighted in yellow!

Com 1 Table 3: Normally pectin contains no protein. The protein content of the oat base without pectin was lower than that of the oat base with pectin, why?

Ans1   It is clear to us that pectin does not contain any protein constituents.

We have observed a slight increase in the amount of plant protein in the plant base after repeating the experiments several times with different plant sources. We do not believe that the amount of protein increases in the presence of pectin as there is no statistically significant difference. We believe that the amount of protein changes insignificantly. The fluctuation is probably due to the interaction of the components of the oat base.

We asked ourselves this question also from a methodological point of view. Taking into account the specificity of our protein determination method, we came to the conclusion that the available free protein is measured to a greater extent in the oat base.

The addition of pectin to the oat base leads to an increase in the measured protein concentration of 0.2-0.3%. We interpret this as an increase in the availability of the measured protein, which can be released from the matrix due to the surfactant properties of pectin. The surface-active properties of pectin have been extensively described: [J.S. Yang, T.H. Mu, M.M. Ma. Extraction, structure, and emulsifying properties of pectin from potato pulp. Food Chemistry, 244 (2018), pp. 197-205;     Li Z, Xi J, Chen H, Chen W, Chen W, Zhong Q, Zhang M. Effect of glycosylation with apple pectin, citrus pectin, mango pectin and sugar beet pectin on the physicochemical, interfacial and emulsifying properties of coconut protein isolate. Food Res Int. 2022 156:111363. doi: 10.1016/j.foodres.2022.111363]

In addition, the acidity (pH) of the oat base decreased as a result of the pectin application (Fig. 2A), which may be the reason for the increase in protein availability for measurement.

Comments and answers on the Quality of English Language:

Line 131: The analysis of the content of protein, fat, solids and total sugars in the samples were determined by near infrared spectroscopy: was - Corrected

Line 187: step - Corrected

Line 198: OH-free radical scavenging ability was carried out following the procedure described by Sungatullina et al. [41]. - Corrected

Line 209: Oat-based products - Corrected

Line 337: As fermented milk, including plant fermented products, are characterized by a number of specific properties: Fermented milk, including plant-fermented products, is characterized by several specific properties - Corrected

Lines 422-423: Oat milk contains higher levels of vitamins and minerals, such as vitamin E, magnesium and potassium, than cow's milk. - Corrected

Line 428: showed that the use of pectin - Corrected

Line 476: On the other hand - Corrected

Line 487: In agreement with our results, the use - Corrected